# The Virome of ‘Lamon Bean’: Application of MinION Sequencing to Investigate the Virus Population Associated with Symptomatic Beans in the Lamon Area, Italy

**DOI:** 10.3390/plants11060779

**Published:** 2022-03-15

**Authors:** Giulia Tarquini, Marta Martini, Simone Maestri, Giuseppe Firrao, Paolo Ermacora

**Affiliations:** 1Department of Agriculture, Food, Environmental and Animal Sciences, University of Udine, I-33100 Udine, Italy; giulia.tarquini@uniud.it (G.T.); marta.martini@uniud.it (M.M.); giuseppe.firrao@uniud.it (G.F.); 2Department of Biotechnology, University of Verona, I-37134 Verona, Italy; simone.maestri@univr.it

**Keywords:** plant disease, virology, bean common mosaic virus, cucumber mosaic virus, peanut stunt virus, bean yellow mosaic virus

## Abstract

‘Lamon bean’ is a protected geographical indication (PGI) for a product of four varieties of bean (*Phaseolus vulgaris* L.) grown in a specific area of production, which is located in the Belluno district, Veneto region (N.E. of Italy). In the last decade, the ‘Lamon bean’ has been threatened by severe virus epidemics that have compromised its profitability. In this work, the full virome of seven bean samples showing different foliar symptoms was obtained by MinION sequencing. Evidence that emerged from sequencing was validated through RT-PCR and ELISA in a large number of plants, including different ecotypes of Lamon bean and wild herbaceous hosts that may represent a virus reservoir in the field. Results revealed the presence of bean common mosaic virus (BCMV), cucumber mosaic virus (CMV), peanut stunt virus (PSV), and bean yellow mosaic virus (BYMV), which often occurred as mixed infections. Moreover, both CMV and PSV were reported in association with strain-specific satellite RNAs (satRNAs). In conclusion, this work sheds light on the cause of the severe diseases affecting the ‘Lamon bean’ by exploitation of MinION sequencing.

## 1. Introduction

Beans have been cultivated for more than 500 years in the province of Belluno (Veneto region, Northeast of Italy). The pole bean named Lamon bean (*Phaseolus vulgaris* L.) is an agricultural product with Protected Geographical Indication (PGI), obtained by growing four local varieties: Calonega, Canalino, Spagnol, and Spagnolet. Its qualities are specifically linked to the area of production, which is limited to 18 municipalities in the Belluno area. In 2012, and frequently also in subsequent years, the pole bean in the Lamon area has been subjected to epidemics caused by viruses that compromised its profitability. Viruses represent a serious threat to global food security and sustainability [1]. Virus infections cannot be fought directly, and their efficient control is difficult. The use of virus-free propagation material, severe phytosanitary measures, appropriate cultural practices, and, when available, the use of resistant cultivars represent the only strategies to contain viral disease in plants [2]. Breeding for virus resistance is often considered the most efficient and simplest way to avoid the losses caused by virus diseases [2]. However, classic breeding techniques seem inappropriate for breeding bean ecotypes grown in Lamon, due to the need to preserve the typical traits of an agri-food product subjected to certification. In recent years, field observations reported the presence of infected bean plants that exhibit mild symptoms, suggesting the existence of tolerant genotypes that may have spontaneously developed tolerance due to the strong pressure of virus infection (Ruggero Osler, personal communication). However, this assumption has been never experimentally assessed. In order to provide experimental evidence, which could confirm or deny the possible presence of tolerant genotypes within Lamon bean ecotypes, it was deemed necessary to lay the basis for the establishment of a reliable, fast, and accurate molecular diagnostic tool that allows the discrimination of viral species affecting bean plants in Lamon. To achieve this purpose, a deep characterization of the viral population associated with symptomatic Lamon beans had to be performed.

Previous studies detected the presence of three main viruses in Lamon bean: bean common mosaic virus (BCMV), cucumber mosaic virus (CMV), and bean yellow mosaic virus (BYMV).

BCMV is one of the most destructive, with a global spread of *Potyvirus* naturally infecting *Phaseolus vulgaris* L. [3]. Its genome is a single-stranded, positive-sense RNA ([+]ssRNA), consisting of a continuous open reading frame (ORF) that is translated into a large polyprotein and subsequently cleaved into mature polypeptides [4]. Two distinct types of symptoms may occur in infected plants, such as leaf mosaicism and “black root”, depending on the viral strain, the environmental conditions, and especially the host genotype [3].

BCMV isolates were classified into seven pathotypes and numbered from I to VII based on their ability to overcome resistance mediated by the individual recessive genes (or their combinations) and symptom expression on distinct bean cultivars [3,5,6].

CMV has the broadest host range of any known virus, infecting more than 1000 different plant species [7,8]. This virus is a multicomponent virus, consisting of three genomic single-stranded RNAs with variable sizes singularly encapsidated in icosahedral particles, and two subgenomic RNAs [8,9]. RNAs-1 and -2 encode non-structural proteins involved in viral replication, while RNA-3 encodes the movement protein, and together with the subgenomic RNA 4, the coat protein [9]. Differentiation of CMV strains into three distinct subgroups, such as IA, IB, and II, was proposed by Roossinck et al. [10], considering host range, aphid vectors, serological data, and sequence similarities among different virus strains.

BYMV is a worldwide spread *Potyvirus*, which can naturally infect both domesticated and wild-type species of several monocotyledonous and dicotyledonous families, often causing significant economic crop losses [11]. Similarly to BCMV, BYMV genome consists of a [+]ssRNA of about 9.7 kb, which includes terminal untranslated regions flanked by a single open reading frame [11]. This large polyprotein is processed into at least 10 different proteins with specific functions [12,13]. Phylogenetic studies inferred on CP gene proposed the subdivision of BYMV into seven different groups, based on the natural hosts and geographical distribution [14]. The biggest group is the “general group”, which includes isolates with generalist host-strategies, infecting a wide range of different hosts [14]. The remaining six groups showed specialist host strategies, including BYMV isolates that were originally recovered from a narrow range of host plants. Among these groups, five groups were named as their natural hosts: the monocot, lupin, broad bean, canna, and pea groups. The sixth monotypic group was indicated as “W”, according to the BYMV W-isolate collected from white lupin (*Lupinus albus*). Wylie et al. [14] also suggested that the “general group” could represent the original ancestral group of BYMV, from which different isolates of the other six specialist groups have evolved [15].

For the first time, this study revealed the presence of peanut stunt virus (PSV) in common bean, which is generally reported to infect the Chenopodiaceae, different Leguminosae, and a limited number of Solanaceae [16]. PSV has a tripartite genome, consisting of three positive-sense RNA molecules. RNA-1 and RNA-2 are monocistronics and encode for the replicase complex, whereas RNA-3 is bicistronic and encodes for the movement and the coat proteins [17].

Phylogenetic relationships among different PSV isolates are constantly evolving. Initially, PSV strains were divided into two groups, the Eastern clade (E-type) and Western clade (W-type), considering host range, immunological data, support for satRNA replication, and sequence homologies [18,19]. Currently, five distinct subgroups numbered from I to V have been established, each represented by a specific strain [19]

Considering the wide diversity of species and viral strains that could be detected in Lamon bean, a fully comprehensive and sensitive diagnostic tool, able to explore such a great variability, was highly needed.

For this reason, we took advantage of the advent of High Throughput Sequencing (HTS) approaches, which in the last two decades have allowed great advances and have promoted discovery, diagnostics, and evolutionary studies [20]. Plant virus research has been heavily impacted by the development of HTS techniques for the identification of emerging viruses, genome reconstruction, analysis of population structures, evolution of novel viral strain(s), and much more [21,22,23,24,25,26]. Single-molecule sequencing technologies, often called “third generation sequencing”, provide greater advantages over second-generation approaches, including the production of long read lengths that are easier to map to a reference sequence and that facilitate de novo assembly, short run times, small amounts of input nucleic acids (DNA or RNA), and low cost for a single run [20]. The consistent improvement in HTS technologies has occurred in parallel with the continuous development of suitable bioinformatic tools for data analysis and with the online availability of a wide range of biological data sets, which have allowed a deeper comprehension of HTS results [23,24].

In this work, a third-generation sequencing platform, namely MinION, provided by Oxford Nanopore Technologies, was exploited to obtain the virome of seven bean samples that exhibited different foliar symptoms resembling those of virus disease. The picture that emerged from the sequencing results was consolidated through RT-PCR and ELISA assays in many samples, inclusive of different varieties of bean grown in the Lamon area and various families of herbaceous plants, which may represent a virus reservoir in the field.

## 2. Results

### 2.1. Virome Determination

The virome of seven Lamon bean plants (varieties Calonega, Spagnolet, and Canalino) which showed either mild (Figure 1A) or severe (Figure 1B) leaf mottling and deformation, as well as mild (Figure 1C) or severe (Figure 1D) extended chlorosis, was obtained by MinION sequencing. MinION sequencing yielded a total of 6,911,238 reads. Of those, 3,918,774 (56.7%) reads were obtained after demultiplexing and quality filtering steps. The selected reads were aligned to a viral sequences database and 51,037 (1.3%) of them found a hit in the database (Figure 2A). The percentage of viral reads in our samples ranged from 0.07% to 3.83%. Computational analyses of viral reads allowed the identification of multiple virus species occurring in single or mixed viral infections (Figure 2B). Two out of seven plants were infected by CMV (NB08 and NB12) and bean yellow mosaic virus (BYMV, NB06, and NB11), while BCMV was detected in four samples (NB08, NB10, NB11, and NB12). Interestingly, PSV was also reported for the first time in three samples of sequenced Lamon bean (NB05, NB10, and NB12), showing the highest abundance among detected viruses. CMV- and PSV-associated satellite RNAs (satRNAs) were detected in samples NB08 and NB12, respectively. The CMV satRNAs showed high sequence similarity with the T4 (X86415.1), T8A (X86408.1), and IR-WI (JF834526.1) strains, as well as with WL3-satRNAs (X51465.1), pBsat2 RNA (M30592.1), and E-satRNA (M20844.1). The PSV satRNAs showed sequence similarity with the satRNA strain P (P6-sat, Z98197.1). A detailed description of sequencing data is reported in Appendix A.

More in-depth analysis of reads of sample NB11 assigned to BYMV showed the presence of systematic differences compared to BYMV genome sequence available in NCBI (NC003492.1). This observation was confirmed also in a contig obtained by de novo assembling such reads (Appendix A). Blast analysis of the obtained contig had BYMV genome sequence as top hit, with 90.1% alignment identity.

### 2.2. RT-PCR and ELISA Detection

A total of 59 samples were tested by RT-PCR assays to confirm the presence of the viruses detected by MinION sequencing, BCMV, CMV, PSV and BYMV (Table 1).

Three Lamon beans were negative to all tested viruses (9%). Single infections caused by BCMV, and PSV were detected in 39% and 13% of Lamon beans, respectively, while CMV was reported in 21% of bean samples, exclusively in co-presence with either BCMV (9%), PSV (3%), or both (9%). Mixed infection by BCMV and PSV was also reported in 12% of Lamon beans. BYMV was identified, exclusively by MinION sequencing, in two samples (6%) either as a single infection (LB-21, NB06) or in co-presence with BCMV (LB-4, NB11). All herbaceous hosts were negative for the presence of BCMV. CMV was exclusively reported in *Trifolium pratense* L. (Leguminosae), both as a single infection (4%) and in co-presence with PSV (8%). Single infection caused by PSV was detected in *Trifolium pratense* L. (8%), *Chrysanthemum* sp. (4%) and, for the first time, in *Solanum tuberosum* L. (8%). Members of the Amaranthaceae (*Chenopodium album* L., *Amaranthus retroflexus* L., and *Achillea millefolium* L.), Apiaceae (*Daucus carota* L.), and Asteraceae (*Taraxacum officinale* L.) were negative to all tested viruses (68%).

The results of ELISA tests are reported in Table 2 and revealed BCMV presence in samples collected in all the surveyed municipalities, with an incidence ranging from 40% to 100%. CMV was serologically identified in 50% and 62% of the samples belonging to the Fonzaso and Sovramonte area, while PSV was found to be present in two samples from two municipalities (Sovramonte and Belluno) with incidence of 25 and 30%, respectively.

### 2.3. BLAST Analysis of Sanger Sequences and Phylogenetic Investigation

Sequences obtained by Sanger were used for assignment to pathotypes/subgroups according to recent comprehensive studies of BCMV [27], CMV [28], and PSV [29]. The results revealed that 79% of BCMV sequences fall into pathotype VII, sharing 97.7–99.2% identity with NL-4 strain (DQ666332) [30]. Three BCMV sequences (13%) were included in the pathotype I, exhibiting from 99.0% to 99.6% of sequence similarity to the NL-1 strain (KM023744) [31]. The remaining BCMV sequences (8%) fall into pathotype VI, sharing 95.6–99.2% identity with recombinant RU-1 (GQ219793) [32] and RU-1-CA (MH024843) [33] strains. All PSV sequences belonged to subgroup IA, showing a sequence similarity that ranged from 92.7% to 99.3% with the PSV-ER strain (U15730) [34]. CMV sequences can be assigned to the subgroups IA and IB, exhibiting 99.3% identity with Fny-CMV strain (D10538) and 95.3–96.4% sequence similarity with As-CMV strain (AF013291), respectively [10,35,36]. Results of BLAST analyses are summarized in Table 3. Phylogenetic analyses (Figure 3) assigned reference sequences of the isolates of BCMV, PSV and CMV detected in Lamon bean plants to the respective clusters congruently with the BLAST results outlined above. The phylogenetic analysis carried out with the introduction of other sequence variants found in the survey produced similar results (not shown).

Moreover, to assess confidence on MinION sequencing as a diagnostic tool for laboratory routine diagnosis, as well as on its reliability in the identification of novel viral strains, Sanger sequences assigned to BCMV, PSV, and CMV were compared to HTS contigs obtained by *de novo* assembly of reads assigned to corresponding viruses by our bioinformatic analysis. Alignments revealed that BCMV (Appendix A), PSV (Appendix A), and CMV (Appendix A) contigs obtained assembling MinION reads showed 100%, 99.5%, and 99.1% sequence similarity with the corresponding Sanger sequences, respectively.

## 3. Discussion

An all-encompassing description of the virome associated with a crop and the availability of high-throughput diagnostic tools are useful to improve viral disease control strategies. The continuous development of newer and even more robust and user-friendly High Throughput Sequencing (HTS) technologies for detecting and identifying viruses makes these new approaches suitable for routine use in testing laboratories, representing a powerful new high-throughput diagnostic tool [20,37].

In this work, MinION sequencing by Oxford Nanopore Technologies was exploited as a diagnostic method for detection and identification of viruses infecting Lamon bean, an agricultural product with protected geographical indication.

MinION sequencing provides many advantages over the widely used RNA-seq approach. Minion platform produces long read lengths, which are easier to map to a reference genome, and facilitate de novo assembly. In addition, this approach significantly reduces run times, requires only a small amount of total RNA, and has low costs for a single run. The MinION approach was preferred to the commonly used RNA-seq, because it can be easily performed in every laboratory using the simple and small sequencer provided with the kit, and data analysis for diagnostic purposes does not necessitate deep knowledge of bioinformatics tools [20].

Overall, a single MinION sequencing run allowed a snapshot of the viromes of seven samples, enabling a high-throughput screening of their infection status. Moreover, since the samples were ligated without the use of PCR, the number of viral reads assigned to each sample can be directly linked to the viral load. Indeed, the percentage of viral reads in our samples suggested a wide range of viral loads.

The virome analysis of seven bean samples, which exhibited different foliar symptoms, revealed the main presence of BCMV, CMV, PSV, and BYMV, which occurred both singly and in mixed infection [38,39,40,41].

RT-PCR assays detected BCMV in many bean samples (75%), whereas it was never reported in herbaceous hosts, thus excluding their possible roles as virus reservoirs in the field.

Sequence analyses and phylogenetic investigations performed on a 505-bp fragment of BCMV polyprotein revealed that the strains found in the Lamon area fall into pathotype I and VII, showing high similarity with the NL1 (KM023744.1) and NL4 (DQ666332.1) strains, respectively [30,31]. Two BCMV sequences have been included in pathotype VI, exhibiting significant sequence similarity to RU-1 (GQ219793) and RU-1-CA (MH024843), which have been reported as capable of overcoming *bc−2^2^*, the most advanced resistance gene in the common bean [32,33,42].

Our data revealed a wide intra-specific genetic variability in the BCMV population that led to a challenging visual assessment of symptoms in infected plants, thus complicating the identification of putative tolerant genotypes. The presence of BCMV isolates that showed high sequence identity with the RU-1 and RU-1-CA strains suggests that the phenomenon of overcoming resistance may also occur [32,33]. Moreover, the presence of mixed infections with different viral species further interferes with evaluation of the type and severity of disease symptoms.

Molecular diagnosis revealed the presence of the cucumovirus CMV, and its associated satellite RNAs (satRNAs) in 25% of the Lamon beans and 7% of the herbaceous hosts tested. The virus was often detected in co-presence with BCMV.

Sequence analysis and phylogenetic studies performed on a 584-bp fragment encoding for the coat protein (CP) gene of CMV demonstrated that viral isolates infecting bean in the Lamon area are genetically similar to the Fny and Cs strains, clustering into subgroups IA and IB, respectively [43]. The CMV-associated satRNAs identified in Lamon bean showed high sequence similarity to the T4, T8A and IR-WI classes, as well as the WL3-satRNAs, pBsat2 RNA, and E-satRNA.

In viral diseases, satRNAs play pivotal roles in symptom expression, by worsening, attenuating, or modifying symptoms in infected plants through different molecular mechanisms [44,45]. The common effect of satRNAs is the attenuation of CMV-related symptoms, which occurs often (but not always) through a noticeable decrease in accumulation of viral RNAs [46,47]. This effect is due to the competition between virus and satRNAs for shared replicase-related factors [47,48].

Different classes of CMV-satRNAs can produce distinct phenotypes on infected plants, depending on both the host and the helper virus [49]. Moreover, the presence of certain satRNAs may also affected the pathogenicity of CMV isolates, depending on whether the disease symptoms are unaffected, exacerbated, or attenuated by the presence of satRNA [50].

The classes of satRNAs detected in Lamon beans have been suggested to attenuate CMV-related symptoms [45,51,52,53], despite there still being a lack of experimental evidence.

The synergy between CMV and potyviruses, such as BCMV, has been well-characterized, mostly (but not exclusively) in cucurbit hosts [38]. Wang et al. demonstrated that in CMV-infected plants, symptom attenuation mediated by CMV-satRNAs was suppressed in the presence of potyviruses such as BCMV [38].

Thus, the co-presence of BCMV and CMV, which has been frequently reported in the Lamon area, could further interfere with symptom expression. Indeed, we can speculate that the presence of BCMV in CMV-infected bean could suppress the beneficial effect of CMV-satRNAs, determining an increase in the CMV concentration in infected tissues, and resulting in worse disease symptoms.

PSV was reported in 33% of the Lamon bean samples, representing the first report of the virus in *Phaseolus vulgaris* L. in Italy. The presence of this virus in Lamon bean was also confirmed by ELISA. PSV was also detected in 22% of samples from herbaceous hosts.

The viral strains detected in Lamon can be ascribed to subgroup I, showing high sequence identity with the PSV-J strain and PSV-ER strain. The PSV-associated satRNA showed significant sequence similarity with the P-satRNA strain, which was reported to be involved in worsening of disease symptoms in PSV-infected plants [54]. As demonstrated for CMV satRNAs, PSV satRNAs are also competitors for the same helper-virus replication machinery, thus determining a decrease in viral concentration in infected tissues [55]. Plant proteome changes in response to the presence of P-satRNA during PSV infection have been investigated, demonstrating that P-satRNA induces significant decrease in levels of proteins involved in carbohydrate metabolism, protein biosynthesis, and stress-related factors, including aminopeptidase protein, which is the satRNA-responsive factor [56].

Interestingly, for the first time PSV has been reported in two plants of *Solanum tuberosum* cultivated nearby a bean patch in the Lamon municipality. PSV isolates collected from *S. tuberosum* were sequenced by Sanger technology, revealing a significant sequence identity (>97.0%) with PSV-ER type strain, clustering into subgroup IA together with the isolate PSV_LB35 (accession OL875025) recovered from Lamon bean plants (data not shown). Further studies including a great number of samples need to be performed to evaluate genetic variability and viral pathogenesis on this novel host.

Our analysis based on MinION sequencing showed that BYMV was detected in two of our samples, but this result was not confirmed by RT-PCR. Two different primer pairs were employed in conventional PCR: the first primer pair was retrieved from the literature [57], while the second one was newly designed for this study. However, neither of these primer pairs were able to detect BYMV in the Lamon beans analysed in this study. We investigated why this may have occurred and discovered that the sequenced strains shared only approximately 90% sequence identity with the BYMV reference sequence. This observation may suggest the appearance of new viral strain(s) [26,41], excluding high sequencing error rate, often ascribed to the MinION technology.

In the past, the high sequencing error rate has been frequently reported for MinION technology, representing a significant weakness of this strategy [58]. Recently, this drawback has been largely mitigated, thanks to the development of new sequencing chemistries and a new generation of highly accurate base-callers, such as Guppy [59].

Given the high proportion of MinION detected viruses validated with gold-standard methods, this novel protocol represents a reliable approach for the rapid and high-throughput detection of viruses in a large number of samples, including different ecotypes of Lamon bean and various herbaceous hosts.

This aspect was further corroborated by comparing Sanger sequences of BCMV, PSV, and CMV with the corresponding sequences obtained through HTS approach.

Overall, our results suggested a wide inter- and intra-specific variability among viruses detected in Lamon beans, revealing the existence of mixed infections that may promote viral interactions, such as synergisms or antagonisms [60,61].

Moreover, the survey is also complicated by the presence of CMV- and PSV-satRNAs, for which confirmation of their involvement in plant responses and symptom expression needs to be done.

The results obtained in this work represent a fundamental approach aimed at describing the viruses involved in the Lamon bean pathosystem. In a context of sustainable management of Lamon bean IGP production, a strategy of selection under epidemic pressure of individuals showing tolerance to viruses may be worthy of consideration. Moreover, the comprehensive knowledge of viruses affecting Lamon beans will help to improve the management strategies based on virus-free propagation material and on chemical control of aphid vectors.

## 4. Materials and Methods

### 4.1. Plant Material

Throughout the 2019 growing season, 16 bean fields sited in six municipalities of the Lamon bean area (Belluno, Feltre, Fonzaso, Lamon, Sovramonte, Trichiana) were monitored for the presence of virus-like symptoms. A total of 59 samples were collected during July 2019. Sampled plants were selected among the symptomatic ones to represent the widest range of symptoms. Among them, 33 of the samples represented different bean varieties cultivated in the Lamon area (Spagnol, Spagnolet, Calonega, and Canalino), while 26 samples included diverse families of herbaceous cultivated or wild plants such as the Leguminosae (*Trifolium pratense* L. and *Medicago sativa* L.), Amaranthaceae (*Chenopodium album* L., *Amaranthus retroflexus* L. and *Achillea millefolium* L.), Apiaceae (*Daucus carota* L.), Solanaceae (*Solanum tuberosum* L.), and Asteraceae (*Taraxacum officinale* (L.) Weber and *Chrysanthemum* sp.). Moreover, at the end of August 2019, a total of 112 bean samples were collected from the above-mentioned areas for serological analyses. In all cases, two fully developed leaves were collected from each plant. Immediately after collection, samples were maintained at 4 °C in a refrigerated box and processed before freezing for ELISA or stored at −80 °C until RNA extraction. MinION sequencing by Oxford Nanopore Technologies was performed on seven Lamon bean samples that exhibited different foliar symptoms, belonging to varieties Calonega, Spagnolet, and Canalino.

### 4.2. RNA Extraction and cDNA Synthesis

Total RNA was extracted from 100 mg of leaf tissues, using a Spectrum^TM^ Plant Total RNA Kit (SIGMA Aldrich, St. Louis, MO, USA) and following the kit manufacturer’s instructions. Five µg of total RNA were reverse transcribed using a Maxima^TM^ H Minus Double-Stranded cDNA Synthesis Kit (Thermo Fisher Scientific, Waltham, MA USA) according to the manufacturer’s protocol. Before library preparation, cDNAs were purified using a GeneJET PCR Purification Kit (Thermo Fisher Scientific, Waltham, MA USA) following the protocol provided.

### 4.3. cDNA Library Preparation

The library was prepared using a Ligation Sequencing Kit in combination with a Native Barcoding DNA Kit (Oxford Nanopore Technologies, Oxford, UK; SQK-LSK109 and EXP-NBD104). The purified cDNA was treated with a NEBNext FFPE DNA Repair and NEBNext Ultra II End-Repair/dA-tailing module (New England Biolabs, Ipswich, MA, USA) according to the protocol provided by the kit (Oxford Nanopore Technologies; SQK-LSK109). The dA-tailed cDNA was cleaned using 1 volume of AMPure XP beads (Beckman Coulter, Indianapolis, IN, USA) and eluted in 25 μL of nuclease-free water. For barcoding cDNA samples, 2.5 μL of one of the barcodes from a Native Barcoding DNA kit (NB05-NB12; Oxford Nanopore Technologies; EXP-NBD104, Oxford, UK) was ligated to the end-prepped cDNAs with 25 μL of Blunt/TA ligase master mix (New England Biolabs). Following bead-purification, the barcoded ligated cDNA was eluted in 26 μL of nuclease free-water. After cDNA quantification with a Qubit fluorometer (Thermo Fisher Scientific, Waltham, MA, USA), barcoded libraries were equimolar-pooled prior to the adapter step. Five μL of adapter mix II (AMII, SQK-LSK109; Oxford Nanopore Technologies) were ligated to 65 μL of pooled barcoded cDNA using 10 μL of Quick T4 DNA ligase (New England Biolabs) and 20 μL of NEBNext Quick Ligation Reaction Buffer (New England Biolabs). The final library was purified with 0.5 volume of AMPure XP beads (Beckman Coulter), eluted in 15 μL of Nanopore Elution Buffer and quantified using the Qubit to estimate the total amount of cDNA prior to sequencing. The library was kept on ice until ready to load on the MinION device.

### 4.4. MinION Sequencing

Sequencing was carried out on the MinION Mk1B platform using R9.4.1 flow cells (FLO-MIN106) (Oxford Nanopore Technologies). The priming of the flow cell was performed according to the manufacturer’s protocol. Twelve microlitres of the final library were mixed with loading beads (LB; Oxford Nanopore Technologies) and sequencing buffer (SQB; Oxford Nanopore Technologies) and loaded onto the SpotON port of the flow cell.

### 4.5. Data Acquisition

After sequencing, data were collected as fast5 files. Base-calling of raw fast5 files was carried out using Guppy v4.2.2 with high-accuracy mode (Appendix A). Reads were demultiplexed with Guppy v4.2.2 requiring the presence of indexes at both ends. Reads were then filtered by quality with NanoFilt v2.7.1 [62], requiring a minimum quality score of 7, and converted to fasta format with seqtk seq (https://github.com/lh3/seqtk accessed 23 June 2021). Reads from each sample were split into smaller files and processed using Parallel [63]. In particular, reads were aligned to the NCBI RefSeq Viral genome sequence database (ftp://ftp.ncbi.nlm.nih.gov/genomes/refseq/viral accessed 23 June 2021) using Nucleotide-Nucleotide BLAST 2.9.0 + [64]. Up to one top hit for each read was retained in case the alignment identity and query coverage were higher than 85% and 80%, respectively. Filtered Blast hits from each file were then merged and a summary file was created containing the number of reads assigned to each taxon, together with the average alignment identity and query coverage. Finally, the R package *taxize* was used to retrieve the full taxonomy for each taxon [65]. Taxonomy bar-plots were obtained using the R packages *pheatmap* and *ggplot2* [66]. Scripts used for bioinformatic analyses are reported in the https://github.com/MaestSi/ONT_preprocessing and https://github.com/MaestSi/MetaBlast (accessed 23 June 2021) [67].

Reads from NB11 sample assigned to BYMV by Blast analysis were then extracted and aligned to BYMV genome sequence from NCBI (NC003492) with command “minimap2 -ax map-ont BYMV_NC003492.fasta NB11_BYMV.fastq | samtools view -hSb | samtools sort -o NB11_BYMV_reads_mapped_to_BYMV_NC003492.bam”. Moreover, reads were de novo assembled with Canu v2.1.1 with command “canu -p BYMV -d BYMV_Canu -genomeSize = 1k -minInputCoverage = 1 -stopOnLowCoverage = 1 -nanopore-raw NB11_BYMV.fastq” and the resulting contig was aligned to BYMV genome sequence as described above. The same procedure was followed for reads from NB12 sample assigned to BCMV, CMV, and PSV viruses, but using the corresponding Sanger sequence as a reference.

### 4.6. Validation of Sequencing Data by RT-PCR

RT-PCR was carried out on 59 samples, including 33 beans and 26 herbaceous hosts. cDNA was obtained from 100 ng total RNA using the recombinant *Moloney Murine Leukemia virus* (MML-V) reverse transcriptase (Promega Corporation, Madison, WI, USA) and a blend of random hexamer primers (Roche Diagnostic, Indianapolis, IN, USA), as described by [68,69].

Plants were tested for the presence of the viruses identified by MinION sequencing, such as bean common mosaic virus (BCMV), cucumber mosaic virus (CMV), peanut stunt virus (PSV), and bean yellow mosaic virus (BYMV), using the primers listed in Table 4. The new set of primers used in this study were designed based on conserved regions of the sequences of BCMV, CMV, PSV, and BYMV strains available in Genbank database.

Virus reference strains kindly provided by Marina Ciuffo from the Institute for Sustainable Plant Protection (IPSP), Turin, Italy were included in each PCR assay with specific primers.

The PCR reaction mixture contained 10 μL of 5X Green GoTaq Flexi buffer (Promega, Madison, WI, USA), 400 μM dNTP mix (Promega, Madison, WI, USA), 1.5 mM MgCl_2_ (Promega, Madison, WI, USA), 0.4 μM each primer, and 1.25 U of GoTaq Flexi DNA Polymerase (Promega, Madison, WI, USA). Nuclease-free water and 2 μL of cDNA were added for a total volume of 50 μL. Amplification cycles were performed according with the following thermal protocol: 95 °C for 3 min; 45 cycles of denaturation at 95 °C for 40 s; 30 s of annealing at either 55 °C (for BCMV, PSV, and BYMV) or 60 °C (for CMV); and extension at 72 °C for 1 min. A final extension at 72 °C for 5 min was also performed. All PCR reactions included a non-template control and a positive control in the form of cDNA from an infected plant. PCR products were analysed with electrophoresis, using 1% TAE agarose gels stained with SYBR Safe DNA gel stain (Thermo Fisher Scientific).

### 4.7. BLAST Analyses of Sanger Sequences and Phylogenetic Investigation

RT-PCR products of Lamon bean samples that tested positive for the presence of BCMV (23 samples), PSV (12 samples), and CMV (7 samples) were sequenced through Sanger technology. Amplicons were purified using a Thermo Scientific GeneJET PCR Purification Kit (Thermo Fisher Scientific) according to the manufacturer’s protocol and sequenced through Sanger technology by BMR Genomics (http://www.bmr-genomics.it/ accessed 7 November 2020). Nucleotide sequences were analysed with Blast sequence analysis tools [64].

To compare with previous phylogenetic works in the literature, reference sequences of strains detected in this investigation were aligned and analyzed with those included in the most recent phylogenetic studies on BCMV [27], CMV [28] and PSV [29]. Phylogeny of BCMV, PSV and CMV isolates infecting Lamon bean plants was reconstructed by adopting the maximum likelihood (ML) method as implemented in Geneious—R10 (Biomatters Ltd., Auckland, New Zealand). To test the robustness of each branch, a total of 500 bootstrap replicates were performed. Sanger sequences were also aligned to corresponding HTS contigs with Blastn [64] to obtain alignment identity values.

### 4.8. Serological Detection

The presence of BCMV, CMV, and PSV was confirmed on a total of 112 samples of Lamon beans through DAS-ELISA assay, using Pathoscreen^®^ kits (Agdia EMEA, Soisy-sur-Seine, France) in accordance with the manufacturer’s protocol. Both commercial positive and negative controls for the above-mentioned viruses were included in each test. After substrate addition, optical density values at 405 nm were measured with a Tecan Spark microplate reader after 30, 60, and 120 min. Samples with absorbance values greater than three times the mean absorbance of negative controls were considered positive.

## Figures and Tables

**Figure 1 plants-11-00779-f001:**
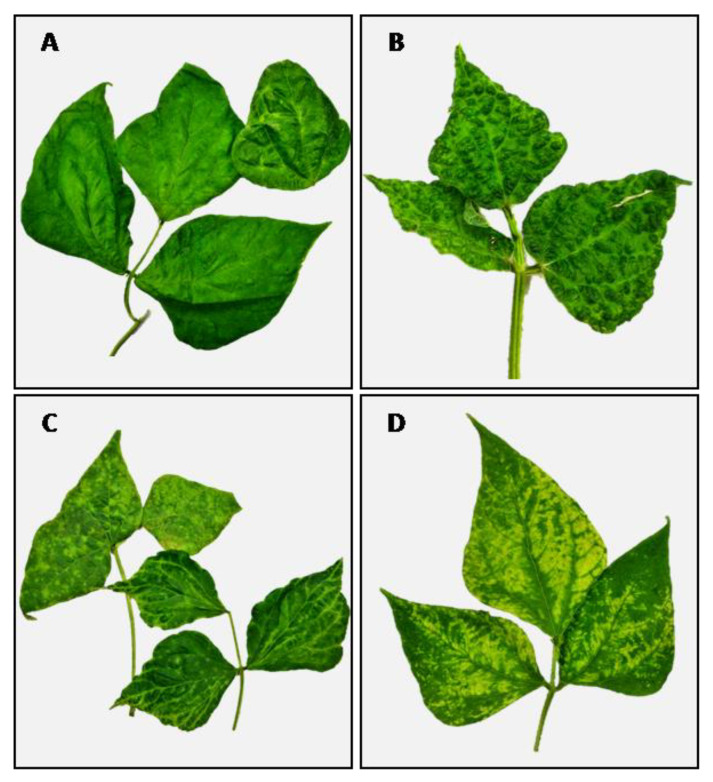
Description of virus-like symptoms observed in Lamon beans. Mild (**A**) and severe (**B**) leaf mottling and deformation on Lamon bean ecotype Canalino and Calonega, respectively. Mild (**C**) and severe (**D**) yellow chlorosis on Lamon bean ecotype Spagnolet.

**Figure 2 plants-11-00779-f002:**
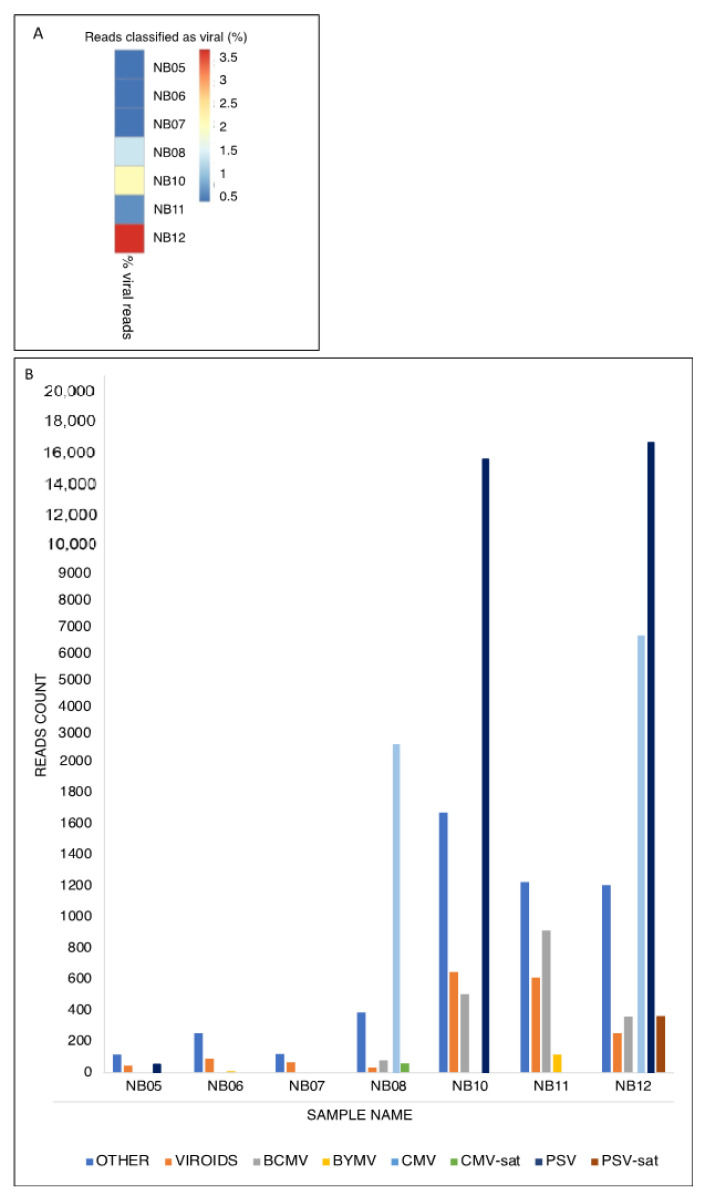
(**A**) Proportion of viral reads produced by HTS sequencing (Oxford Nanopore Technologies). For each sample, the proportion of reads with a hit in the viral database is reported; (**B**) Virome composition. Bars represent relative abundance of viral species for each sample. Only reads with a hit in the viral database are considered. BCMV = bean common mosaic virus; BYMV = bean yellow mosaic virus; CMV = cucumber mosaic virus; CMV sat. RNA = cucumber mosaic virus satellite RNA; PSV = peanut stunt virus; PSV sat. RNA = peanut stunt virus satellite RNA.

**Figure 3 plants-11-00779-f003:**
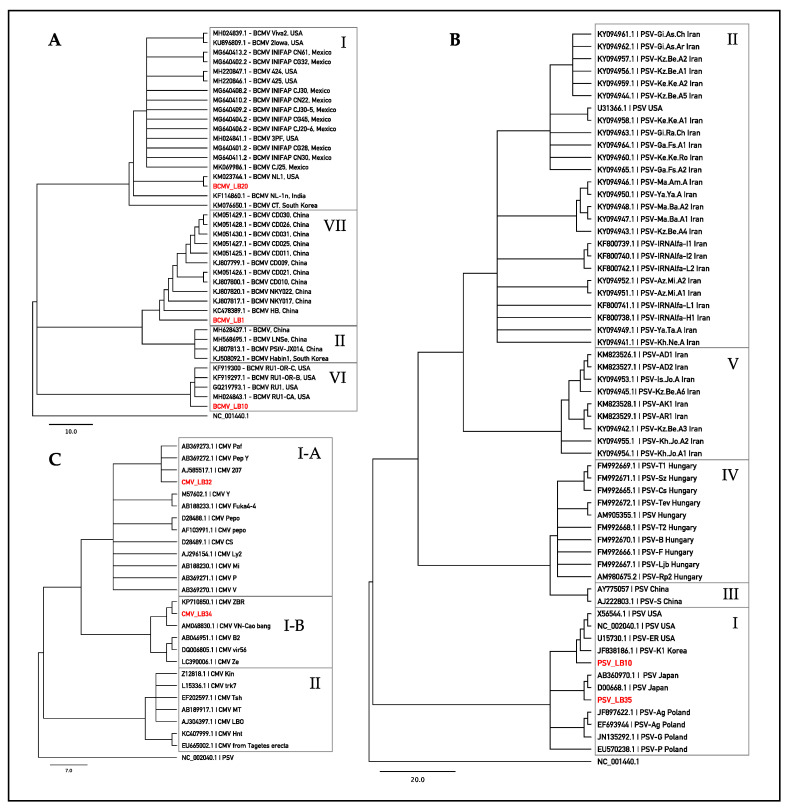
Phylogenetic trees of BCMV (**A**), CMV (**B**) and PSV (**C**) isolates infecting Lamon bean. Trees were constructed by the Maximum-Likelyhood (ML) method implemented in Geneious R-10 software (Biomatters Ltd.). Branch support was computed using the bootstrap method based on 500 replicates. Cucumber mosaic virus (CMV, NC_001440.1) was used as outgroup for BCMV and PSV viruses, while peanut stunt virus (PSV, NC_002040.1) was used as outgroup for CMV virus.

**Table 1 plants-11-00779-t001:** Results of RT-PCR detection of viruses in Lamon bean and herbaceous hosts collected in different municipalities.

Family	Species	Variety	Sample ID	Site	RT-PCR			
					BCMV	CMV	PSV	BYMV
Leguminosae	*P. vulgaris*	Unknown	LB-8	Fonzaso	-	-	+	-
		Calonega	LB-2	Feltre	+	-	+	-
			LB-10	Feltre	+	-	+	-
			LB-11	Trichiana	-	-	-	-
			LB-15	Trichiana	+	-	-	-
			LB-19	Feltre (NB05)	-	-	+	-
			LB-25	Feltre	-	-	-	-
			LB-30	Sovramonte	+	-	-	-
			LB-35	Sovramonte	+	+	+	-
			LB-36	Sovramonte	+	+	-	-
		Canalino	LB-1	Feltre	+	-	-	-
			LB-7	Fonzaso	+	-	-	-
			LB-18	Feltre (NB10)	+	-	+	-
			LB-24	Fonzaso (NB12)	+	+	+	-
			LB-26	Sovramonte	+	+	+	-
			LB-27	Sovramonte	+	-	-	-
		Spagnol	LB-32	Lamon	-	+	+	-
			LB-9	Feltre	+	-	-	-
			LB-34	Sovramonte	+	+	-	-
		Spagnolet	LB-3	Feltre	+	-	-	-
			LB-12	Trichiana	-	-	+	-
			LB-13	Trichiana	+	-	-	-
			LB-14	Trichiana	+	-	-	-
			LB-16	Feltre	+	-	-	-
			LB-17	Feltre	+	-	-	-
			LB-22	Feltre (NB07)	-	-	-	-
			LB-23	Feltre	-	-	+	-
			LB-33	Feltre	+	-	+	-
			LB-4	Feltre (NB11)	+	-	-	-
			LB-5	Feltre	+	-	-	-
			LB-6	Feltre (NB08)	+	+	-	-
			LB-20	Feltre	+	-	-	-
			LB-21	Feltre (NB06)	-	-	-	-
	*Trifolium pratense* L.	-	HO-21	Feltre	-	+	+	-
			HO-25	Feltre	-	+	+	-
			HO-34	Fonzaso	-	-	+	-
			HO-35	Fonzaso	-	-	+	-
			HO-36	Fonzaso	-	-	-	-
			HO-37	Fonzaso	-	-	-	-
			HO-48	Sovramonte	-	+	-	-
	*Medicago sativa* L.	-	HO-24	Feltre	-	-	-	-
Amaranthaceae	*Chenopodium album* L.	-	HO-2	Trichiana	-	-	-	-
			HO-39	Fonzaso	-	-	-	-
	*Amaranthus retroflexus* L.	-	HO-38	Fonzaso	-	-	-	-
	*Achillea millefolium* L.	-	HO-41	Lamon	-	-	-	-
Apiaceae	*Daucus carota* L.	-	HO-47	Sovramonte	-	-	-	-
Solanaceae	*Solanum tuberosum* L.	-	HO-23	Feltre	-	-	+	-
			HO-46	Lamon	-	-	+	-
Asteraceae	*Taraxacum officinale* L.	-	HO-12	Feltre	-	-	-	-
			HO-13	Feltre	-	-	-	-
			HO-14	Feltre	-	-	-	-
			HO-15	Feltre	-	-	-	-
Asteraceae	*Taraxacum officinale* L.	-	HO-49	Sovramonte	-	-	-	-
			HO-50	Sovramonte	-	-	-	-
Asteraceae	*Weber and Chrysanthemum* sp.	-	HO-22	Feltre	-	-	-	-
			HO-42	Lamon	-	-	-	-
			HO-43	Lamon	-	-	-	-
			HO-44	Lamon	-	-	+	-
			HO-45	Lamon	-	-	-	-

**Table 2 plants-11-00779-t002:** Sampling location, and infection rate of BCMV, CMV and PSV determined by ELISA in Lamon bean samples collected in August 2019.

Municipality	N° of Samples	ELISA Based Incidence (%)					
		BCMV	CMV	PSV	Mixed Infections Rate		
					BCMV and CMV	BCMV and PSV	BCMV, CMV and PSV
Belluno	20	40	0	30	0	30	0
Feltre	36	72	0	0	0	0	0
Fonzaso	4	100	50	0	50	0	0
Lamon	32	81	0	0	0	0	0
Sovramonte	16	100	62	25	62	25	12.5
Trichiana	4	100	0	0	0	0	0

**Table 3 plants-11-00779-t003:** BLAST analysis of viral sequences obtained by RT-PCR from Lamon bean samples.

Sample ID	Variety	Sequence Name	Accession	Strain *	Sequence Similarity (%)	Pathotype or Subgroup
LB-1	*Canalino*	BCMV_LB1	OL874991	NL4	99.0	VII
LB-2	*Calonega*	BCMV_LB2	OL874987	NL4	98.8	VII
		PSV_LB2	OL875028	PSV-ER	97.8	IA
LB-3	*Spagnolet*	BCMV_LB3	OL875007	NL4	99.2	VII
LB-4	*Spagnolet*	BCMV_LB4	OL874992	NL4	99.0	VII
LB-5	*Spagnolet*	BCMV_LB5	OL874997	NL4	99.2	VII
LB-6	*Spagnolet*	BCMV_LB6	OL874986	NL4	98.7	VII
		CMV_LB6	OL875010	Fny-CMV	99.2	IA
LB-7	*Canalino*	BCMV_LB7	OL874998	NL4	99.2	VII
LB-9	*Spagnol*	BCMV_LB9	OL874999	NL1	99.2	I
LB-10	*Calonega*	BCMV_LB10	OL874988	RU-1	99.2	VI
		PSV_LB10	OL875020	PSV-ER	92.7	IA
LB-12	*Spagnolet*	PSV_LB12	OL875026	PSV-ER	97.8	IA
LB-13	*Spagnolet*	BCMV_LB13	OL875000	NL4	99.0	VII
LB-14	*Spagnolet*	BCMV_LB14	OL874995	NL4	99.0	VII
LB-15	*Calonega*	BCMV_LB15	OL874990	NL4	99.0	VII
LB-16	*Spagnolet*	BCMV_LB16	OL875001	NL1	99.2	I
LB-17	*Spagnolet*	BCMV_LB17	OL874994	NL4	97.7	VII
LB-18	*Canalino*	BCMV_LB18	OL875002	NL4	98.8	VII
		PSV_LB18	OL875019	PSV-ER	99.3	IA
LB-19	*Calonega*	PSV_LB19	OL875023	PSV-ER	97.8	IA
LB-20	*Spagnolet*	BCMV_LB20	OL875009	NL1	99.6	I
LB-23	*Spagnolet*	PSV_LB23	OL875027	PSV-ER	97.9	IA
LB-24	*Canalino*	BCMV_LB24	OL875003	NL4	98.8	VII
		CMV_LB24	OL875014	Fny-CMV	99.3	IA
		PSV_LB24	OL875021	PSV-ER	98.9	IA
LB-26	*Canalino*	BCMV_LB26	OL874996	RU1-CA	95.6	VI
		CMV_LB26	OL875015	Fny-CMV	99.3	IA
		PSV_LB26	OL875017	PSV-ER	98.9	IA
LB-27	*Canalino*	BCMV_LB27	OL875008	NL4	99.2	VII
LB-30	*Calonega*	BCMV_LB30	OL875004	NL4	99.2	VII
LB-32	*Spagnol*	CMV_LB32	OL875012	Fny-CMV	99.2	IA
		PSV_LB32	OL875024	PSV-ER	98.7	IA
LB-33	*Spagnolet*	BCMV_LB33	OL875005	NL4	99.2	VII
		PSV_LB33	OL875022	PSV-ER	98.7	IA
LB-34	*Spagnol*	BCMV_LB34	OL874993	NL4	99.2	VII
		CMV_LB34	OL875011	As-CMV	96.6	IB
LB-35	*Calonega*	BCMV_LB35	OL875006	NL4	99.2	VII
		CMV_LB35	OL875013	As-CMV	95.3	IB
		PSV_LB35	OL875025	PSV-ER	99.2	IA
LB-36	*Calonega*	BCMV_LB36	OL874989	NL4	98.8	VII
		CMV_LB36	OL875016	As-CMV	96.4	IB

* Accession numbers of strains: BCMV-NL4 (DQ6663321); BCMV-NL1 (KM023744.1); BCMV-RU1 (GQ219793); BCMV-RU1-CA (MH024843.1); PSV-ER (U15730); Fny-CMV (D10538); As-CMV (AF013291).

**Table 4 plants-11-00779-t004:** List of primers used in conventional PCR for the detection of viruses identified by the HTS approach in Lamon bean plants.

Virus	Primer Sequence (5′ → 3′)	Tm	Size	Reference
Bean common mosaic virus (BCMV)	For: ACCACGCTGCAGCTAAAGAGAACA Rev: AATCTAGATGATATCATACTCTCTA	55 °C	657 bp	Xu and Hampton (1996)
Cucumber mosaic virus (CMV)	For: CAGGTGGTTAACGGTACTTT Rev: CGGTAGAATCAAATTTCGGC	60 °C	748 bp	This study
Peanut stunt virus (PSV)	For: AGCCGTCGATATACCTTTTG Rev: CTCTTCACAATCACCAGGAG G	55 °C	1033 bp	This study
Bean yellow mosaic virus (BYMV)	For: CAGTTTATTATGCAGCGG Rev: GTTACCATCAATCTTCCTGCC	55 °C	644 bp	Uga et al., 2005
Bean yellow mosaic virus (BYMV)	For: TGAAGGGCATTTTGTCAACA Rev: TTAATGAGCTTGCCGTCAAA	55 °C	669 bp	This study

## Data Availability

Raw sequencing data of MinION sequencing have been submitted to NCBI GenBank (BioProject PRJNA805269) under accession numbers SRR18004647-SRR18004653. The nucleotide sequence data of isolates characterized in this study are available in the NCBI GenBank database under accession numbers OL874986-OL875028.

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
