# Peer review of "The Virome of ‘Lamon Bean’: Application of MinION Sequencing to Investigate the Virus Population Associated with Symptomatic Beans in the Lamon Area, Italy"

_plants, 2022, doi:10.3390/plants11060779_

Round 1

Reviewer 1 Report

Comments of the manuscript plants-1599864, entitled "The virome of ‘Lamon bean’: application of MinION Sequencing to investigate the virus population associated with symptomatic beans in the Lamon area, Italy”

This manuscript describes the virome of seven bean symptomatic samples by the MinION sequencing platform, including the detection of those viruses through RT-PCR and ELISA in different cultivated and wild plant species.

This work is interesting and has merit, whereas it appears to identifying a spectrum of viruses affecting Lamon bean crops, with the occurrence confirmation for some of them either in cultivated and wild plant species. However, there is a weakness with the manuscript in its current form.  While authors attempt to claim the incidence of plants showing virus-like symptoms, and this is appropriate, the approach strategy to get those (% values) results is not mentioned to confirm viral diseases. Based on this, and also with the limitation about the potential association between symptoms and virus occurrence, makes the first part of results unclear. I would recommend to drop-off the 2.1 (plant material) from the result section.

Another concern is about the potential comparison between PCR and ELISA results in those producing areas.  Both approaches were nicely executed, and as such, I would rather recommend to change the focus on the paper, and despite of limitations by symptom description and virus detection, I think that these results could be combined into one detailed table, further providing the status of those viruses by area and plant species.

Moreover, considering the genetic characterization of different PCR fragments for each virus isolate, and that Discussion is mainly invoking on aspects about genetic variability of each virus species, it would be highly recommended to analyze the phylogenetic relationship of these isolates including those already in the database. This would provide strength results, and conclusions will be well supported. Otherwise results are very preliminary to support the conclusions, and the claims would be very speculative.

I think that the revision level for this work is major, and would need to edit substantial and appropriate changes into text sections.  

Minor suggestions:

Ln13-14: I would delete the sentence “The emergence of …. leading to …crop production. It is a very general assertion.

Ln16: it might be not necessary to indicate the company of the sequencing platform in the abstract.   

Ln45: it is not clear what and why the Fa.La.Res. project is introduced in this section.

Ln49: Please, clarify the context. It is previously mentioned the existence of tolerant genotypes, and thus, it is not clear the rationale of this work.  

Ln 87: Please, clarify if 3 or 4 varieties. In M&M section, there are 4 varieties. Also good to know the number of plants for each variety.

Ln115: exclusively is duplicated

Ln 119: How many samples?

Ln 126-127: Please put this sentence into the M&M section.

Author Response

REFEREE #1

While authors attempt to claim the incidence of plants showing virus-like symptoms, and this is appropriate, the approach strategy to get those (% values) results is not mentioned to confirm viral diseases. Based on this, and also with the limitation about the potential association between symptoms and virus occurrence, makes the first part of results unclear. I would recommend to drop-off the 2.1 (plant material) from the result section.

R: The primary aim of this work was not to establish an association between symptoms and virus presence, instead the study aimed to provide a detail characterization of virus population in Lamon area with main focus on Lamon bean. Symptoms were considered, exclusively, for the choice of samples to be sequenced by MinION approach, since the costs of the method didn't allow its execution on a large number of samples (59). To perform a reliable/robust study, all type of symptoms observed in different Lamon bean ecotypes cultivated in different Lamon area were considered in the analysis. However, on its current form, description of symptoms may be misinterpreted. The section “plant material” in the Results was removed, according to your suggestion. A briefly description of symptoms (Figure 1) that were considered for the choice of samples to be sequenced with MinION was moved in the new section 2.1 “Virome reconstruction”.

Another concern is about the potential comparison between PCR and ELISA results in those producing areas.  Both approaches were nicely executed, and as such, I would rather recommend to change the focus on the paper, and despite of limitations by symptom description and virus detection, I think that these results could be combined into one detailed table, further providing the status of those viruses by area and plant species.

R: Thank you very much for the positive evaluation on execution of PCR and ELISA approaches. While MinION sequencing and PCR assays aimed to identify the viruses that affect Lamon bean plants, the aim of ELISA test was to confirm with an independent method the virus presence and to provide evidence about the spreading of each single virus in different Lamon areas. Immunological detection assays, such as ELISA, are significantly cheaper than the molecular approaches, allowing to investigate a great number of samples in reduced times.

In this study, the  plants (112) which have been subjected to ELISA test were sampled in a different period compared to those investigated by molecular approaches (59), such as MinION and PCR. Hence, results could not be combined into one table because plants are different.

Moreover, considering the genetic characterization of different PCR fragments for each virus isolate, and that Discussion is mainly invoking on aspects about genetic variability of each virus species, it would be highly recommended to analyze the phylogenetic relationship of these isolates including those already in the database. This would provide strength results, and conclusions will be well supported. Otherwise results are very preliminary to support the conclusions, and the claims would be very speculative.

R: Phylogenetic analyses have been included in the study

Minor suggestions Referee #1:

Ln13-14: You are right. Sentence was deleted according to your advice

Ln16: we corrected according to your suggestion.

Ln45: we changed the text according to your suggestion

Ln49: You are right. Sentence was revised according to your advice

Ln 87: The ecotypes of Lamon bean samples objected of this study are four (as reported in M&M section). However, the seven Lamon bean samples subjected to MinION sequencing belonged to three ecotypes, such as Canalino, Calonega and Spagnolet. Number of plants per ecotypes was indicated in the Table 1.

Ln115: You are right. Repetition was deleted according to your advice

Ln 119: 100% of herbaceous hosts were negative for the presence of BCMV. Sentence was revised according to your advice

Ln 126-127: we changed the text according to your suggestion

Reviewer 2 Report

Dear authors, the manuscript dealing with bean virome diagnosed with MinION sequencing is well written (only few corrections). However, I wonder why you did not integrate the citrus exocortis viroid in your analysis as this viroid is present in the seven symptomatic plants subjected to MinION sequencing and this viroid is known to produce symptoms on citrus including leaf yellowing.

Author Response

REFEREE #2

Dear authors, the manuscript dealing with bean virome diagnosed with MinION sequencing is well written (only few corrections). However, I wonder why you did not integrate the citrus exocortis viroid in your analysis as this viroid is present in the seven symptomatic plants subjected to MinION sequencing and this viroid is known to produce symptoms on citrus including leaf yellowing.

R: Thank you very much for the positive evaluation of our work. At that moment, Citrus exocortis viroid (CEVd) and Citrus exocortis Yucatan viroid (CEYVd) were exclusively detected by MinION sequencing. The presence of these viroids in common bean plants has never been reported and an efficient PCR protocol for their diagnosis has never been established. Preliminary in silico analyses revealed a great  diversity among sequences of  CEVd and CEYVd isolates reported in this study and those available in public database. This diversity could be underpinned of a significant genetic variability, maybe due to host-specificity, or might be related with high error rates occurred during sequencing/basecalling. A detailed study is needed to ascertain the presence of CEVd and CEYVd in bean samples, also providing a detailed analysis of phylogenetic relationships with other isolates reported in the literature and available in public database. This aspect was not a part of the purposes of the current work, and it will be addressed in a future study.

Minor suggestion referee #2:

Ln 97 You are right. The figure was modified according to your revision

Ln 275 At that moment, the presence of CMV- and PSV- satRNAs was exclusively reported by MinION sequencing and need to be investigated by appropriate molecular approaches. The role of satRNA in symptom expression is complex to decipher, mostly in plants subjected to mixed infection, where different satRNA can affect multiple biological functions (e.g., replication and/or movement) of the helper virus though different ways, by increasing or reducing its virulence. The contribution of satRNAs in modulation of symptoms in Lamon bean samples will have to be considered in future studies, since it was not addressed in the current work whose the primary aim was to describe virus population and genetic variability in symptomatic beans cultivated in Lamon area.

Reviewer 3 Report

The study describes a very well performed, analyzed and presented identification of viruses infecting beans in the Lamon area of Italy applying a new platform MinIon sequencing. Methodically, the experiment was properly designed and performed. The subject matter of the work is quite interesting, but discussion part will be improved e.g. addressing the reason for the use of MinIon platform rather than regular RNA-seq, things that simple comparisons them each other. This study quite promising for the bean cultivation. I would suggest some input from the side of the authors regarding progress of results and their importance for the applied agriculture. More issues revised were marked yellow in the attached file.

Author Response

REFEREE #3

The study describes a very well performed, analyzed and presented identification of viruses infecting beans in the Lamon area of Italy applying a new platform MinIon sequencing. Methodically, the experiment was properly designed and performed. The subject matter of the work is quite interesting, but discussion part will be improved e.g. addressing the reason for the use of MinIon platform rather than regular RNA-seq, things that simple comparisons them each other. This study quite promising for the bean cultivation. I would suggest some input from the side of the authors regarding progress of results and their importance for the applied agriculture. More issues revised were marked yellow in the attached file.

R: Thank you very much for the positive evaluation of our work. A brief comparison between MinION and RNA-seq approaches has been included in the discussion (Ln 195 - 202)

Minor suggestion referee #3:

Ln 11 Corrections have made in the text.

Ln 77 Corrections have made in the text.

Fig. 1 Captions: Corrections have made in the caption of all figures.

Lns 88 - 89 6’911’238 are the total reads obtained through MinION sequencing, while 3’918’774 are the reads that have been demultiplexed and had PASS sequencing quality. 51’037 are the read that were aligned to a viral sequences database.

Fig. 2B: You are right. The figure was corrected according to your revision.

Ln 110 Correction have made in the text.

Ln 154 A brief comparison between MinION and RNA-seq approaches has been included in the discussion (Ln 195 - 202)

Ln 159 A typing error occurred. Correction was made in the text.

Lns 170 – 171 and Table 4 The name of the viruses are revised according to your suggestion.

Ln 371 and table 4 Correction have made in the text and in table 4.

Reviewer 4 Report

In the paper entitled ,,The virome of ‘Lamon bean’: application of MinION Sequencing to investigate the virus population associated with symptomatic beans in the Lamon area, Italy” submitted to ,,Plants” under ID plants-1599864 Authors described analysis of virus population infecting bean in the Lamon area (Italy). Authors obtained full virome of seven bean samples using MinION sequencing. Moreover, in 2019 they collected 59 samples comprising 33 samples of different bean varieties and 26 samples of herbaceous cultivated or wild plants that were tested using RT-PCR. The other 112 bean samples collected at the end of August in 2019 were subjected for serological analysis. Results revealed the presence of bean common mosaic virus (BCMV), cucumber mosaic virus (CMV), peanut stunt virus (PSV), and bean Yellow Mosaic Virus (BYMV), which often occurred in mixed infections. Moreover, both CMV and PSV were reported in association with strain-specific satellite RNAs (satRNAs). The paper could be interesting for researchers however many points need to be clarified so my recommendation is major revision.

Please, find my specific comments below:

Line 117 -  herbaceous hosts were negative for the presence of BCMV and BYMV as well? BYMV is not included to the table 1 (line 143-144). Maybe it will good to provide ,,-“ showing negative results in both tables or indicate them somehow.

Line 117 - why the Authors did not design the primers for BYMV detection based on the BYMV sequence obtained using MinION sequencing? please explain

Line 123 – Authors said that members of Asteraceae (Taraxacum officinale L.) were negative to all tested viruses. In Table 1 samples HO-44 is marked to be infected with PSV however it is not clear which sample was collected from which host.

Line 218 – please take this information into account (review by Kouadio et al. 2013; Biotechnol. Agron. Soc. Environ. 2013 17(4), 644-650)

More than 100 satRNAs associated with CMV isolates originating from geographically different countries and host plants, of which most were Solanaceae plants such as tomato and tobacco, have been characterized by sequencing and nucleotide sequences were deposited in the National Center for Biotechnology Information (NCBI) GenBank database. These CMV isolates can be classified into at least three groups depending on whether the disease symptoms are unaffected, exacerbated or attenuated by the presence of satRNA on tomato indicator plants (Collmer et al., 1992; Garcia-Arenal et al., 1999)

Line 288 - Plant viruses can emerge in crops from wild plant hosts in which they are often asymptomatic. In addition, many wild plants appear to have multiple infections, including both acute and persistent viruses. I don´t think that focusing only in symptomatic plants is the right choice for this type of work. It will only give the researchers an idea of the viruses that “explain” symptoms, but it will miss any asymptomatic infection in the field.

Line 295 – please explain why those plants were subjected to ELISA only instead of RT-PCR? It is not clear whether those plants were also tested for BYMV?

Line 354 – did Authors verify the samples of CMV and PSV for the presence of satRNAs by RT-PCR?

Line 337 – high throughput sequencing data should be deposited in Sequence Read Archive (SRA), available through multiple cloud providers and NCBI servers

Author Response

REFEREE #4

Line 117 - herbaceous hosts were negative for the presence of BCMV and BYMV as well? BYMV is not included to the table 1 (line 143-144). Maybe it will good to provide ,,-“ showing negative results in both tables or indicate them somehow.

R: Yes, herbaceous hosts were negative for the presence of BCMV and BYMV. You are right, a column with negative results of detection of BYMV was added in table 1 for both bean samples and herbaceous hosts. Plants that were negative for the presence of tested viruses were indicated as “-“.

Line 117 - why the Authors did not design the primers for BYMV detection based on the BYMV sequence obtained using MinION sequencing? please explain

R: The presence of BYMV was assessed in all samples using both the primers reported in literature and those developed in this study (as reported in Ln 320-321). The latter were constructed on conserved regions of the alignment of BYMV sequences available in public database, including the reference of sequencing data (NC003492.1).  BYMV sequences obtained by MinION sequencing were excluded by the alignment because their low abundance (126 reads in total), which allowed a partial coverage of the alignment, and their scarce quality. However, none of primer pairs allowed the detection of BYMV in our samples. As reported in the discussion (Ln 326-333), this result might rely on the occurrence of high rate of errors occurred during sequencing and/or base calling, or it may indicate the evolution of a novel variant of BYMV in Lamon area. Further studies are needed to investigate these hypotheses.

Line 123 – Authors said that members of Asteraceae (Taraxacum officinale L.) were negative to all tested viruses. In Table 1 samples HO-44 is marked to be infected with PSV however it is not clear which sample was collected from which host.

R: You are right. HO-44 is a Chrysanthemum sp. Correction have made in the table.

Line 218 – please take this information into account (review by Kouadio et al. 2013; Biotechnol. Agron. Soc. Environ. 2013 17(4), 644-650)

“More than 100 satRNAs associated with CMV isolates originating from geographically different countries and host plants, of which most were Solanaceae plants such as tomato and tobacco, have been characterized by sequencing and nucleotide sequences were deposited in the National Center for Biotechnology Information (NCBI) GenBank database. These CMV isolates can be classified into at least three groups depending on whether the disease symptoms are unaffected, exacerbated or attenuated by the presence of satRNA on tomato indicator plants (Collmer et al., 1992; Garcia-Arenal et al., 1999)”

R: Many thanks for the kind suggestion. A brief paragraph about the role of satRNA in taxonomy of CMV, depending on whether the disease symptoms are unaffected, exacerbated or attenuated by the presence of satRNA, has been included in the discussion (Ln 278 – 281) with the suggested citation (38).

Line 288 - We decided to focus the attention on symptomatic beans and also on symptomatic wild plants sampled nearby the bean fields because we thought this was the most reasonable approach to follow in order to get some preliminary epidemiological features.

Line 295 – please explain why those plants were subjected to ELISA only instead of RT-PCR? It is not clear whether those plants were also tested for BYMV?

For the first question we can refer to the response to the Reviewer #1: “While MinION sequencing and PCR assays aimed to identify the viruses that affect Lamon bean plants, the aim of ELISA test was to confirm with an independent method the virus presence and to provide evidence about the spreading of each single virus in different Lamon areas. Immunological detection assays, such as ELISA, are significantly cheaper than the molecular approaches, allowing to investigate a great number of samples in reduced times”.The plants analyzed by ELISA were not tested for BYMV because molecular analyses by RT-PCR did not evidence the presence of this virus.

Line 354 – did Authors verify the samples of CMV and PSV for the presence of satRNAs by RT-PCR?

R: CMV-satRNA was detected in sample NB08, while PSV-satRNA was reported in sample NB12, exclusively through MinION sequencing. The presence of satRNAs was not assessed via RT-PCR, since were not an object of this current study. Their fine characterization as well as their involvement in symptom modulation in Lamon bean plants will be addressed in future studies. 

Line 337 – high throughput sequencing data should be deposited in Sequence Read Archive (SRA), available through multiple cloud providers and NCBI servers

R: Raw sequencing data of MinION sequencing have been submitted to NCBI GenBank (BioProject PRJNA805269) under accession numbers SRR18004647-SRR18004653.

Round 2

Reviewer 1 Report

The authors have adequately addressed my comments, and the modified parts of the text have improved the clarity of the manuscript. I have no further major concerns. Some minor points that could be corrected is below.

Ln43-45. Please, cite this statement.

Ln 49-51. This reasoning behind of the work doesn’t fit with the description of the work performed. I think that this work doesn’t show any evidence of resistant genotypes within beans. So, I would suggest to re-write this reasoning accordingly.  

Ln 364. Worth providing additional information on those isolates of PSV detected for first time in potato plants. Were those included in the sanger sequencing assay? If so, how relate with the rest?

L371. Also worth providing additional info on the novel BYMV isolate undetected by conventional RT-PCR.    

Author Response

Thanks, please see attached

Reviewer 4 Report

Authors revised the paper taking into account my comments. I do not have further questions. 

Author Response

Thank you very much